# Access to Multiple Habitats Improves Welfare: A Case Study of Two Zoo-Housed Black Bears (*Ursus americanus*)

**Kelly Bruno [1],\*, Cassidy Hubbard [2]** and **Emily Lynch [3]**

1    Nicholas School of the Environment, Duke University, Durham, NC 27708, USA
2    Department of Biological Sciences, North Carolina State University, Raleigh, NC 27695, USA
3    The North Carolina Zoo, Asheboro, NC 27205, USA
\*    Correspondence: kelly.a.bruno@duke.edu

**Abstract:** Using various forms of enrichment, animal care specialists encourage species-specific behaviors and discourage stereotypic behaviors. Within the zoo community, bears (*Ursids* spp.) are commonly housed, yet are prone to exhibiting stress-related behaviors. Here, we assess the effect of access to multiple habitats, including areas of off guest view, on the welfare of two American black bears (*U. americanus*) housed at the North Carolina Zoo. In this study, we looked at two behaviors, pacing and foraging to represent negative and positive welfare indicators. We performed logistic regressions to model the effect of access on these behaviors. Because having an animal visible to guests is important to consider when creating management plans, we also explored the effect of access on the bears' visibility. We found that full access reduced the likelihood of pacing by an average of 13% and increased the likelihood of foraging by an average of 5%. Access to multiple areas reduced the probability of visibility by 57% for one individual but did not impact visibility of the other bear. This case study suggests the value of access to zoo animal welfare and should incite future research aimed at exploring the effects of access on various behavioral outcomes.

**Keywords:** zoo; environmental enrichment; choice and control; animal welfare; American black bear; carnivore

## 1. Introduction

Encouraging positive welfare states for animals under human care is a central mission for zoos and aquariums around the world. One method for assessing welfare is through behavioral observation. Generally, positive welfare is associated with the exhibition of species-specific behaviors, while stereotypic behaviors [1,2] are considered an indication of diminished welfare [1,3,4]. Stereotypic behaviors can reduce the amount of time engaged in species-specific behaviors [5,6]. Enrichment, exhibit additions, and alteration in husbandry routines can be used to provide opportunities for choice and control and the mitigation of the occurrence of stereotypy [2,5–8].

Providing choices for an animal within their environment can add complexity and present challenges and opportunities to exhibit species-specific behaviors [9,10]. Automatic feeders to disperse food widely and at set or random intervals [7], ice blocks containing food items [11], or unpredictability in schedules [9,12] have all been employed as forms of environmental enrichment [10] to introduce variety in an animal's daily routine. Another method of providing choice is through offering access to various enclosure areas, including those away from guest view [2,13,14]. Presenting options for space use may reduce stress-related behaviors [13–17]. For example, eastern black-and-white colobus monkeys (*Colobus guereza*) and chimpanzees (*Pan troglodytes*) with access to both indoor and outdoor areas were reported to be more active and interacted with enrichment objects more often, suggesting improved welfare [eastern black-and-white colobus monkeys: [8]; chimpanzees: [18]. In addition, multiple studies have shown that when animals were given free access to more than one enclosure area, signs of behavioral agitation lessened; this

was determined to most likely be due to increased choice rather than increased stimulation from additional areas [giant pandas, *Ailuropoda melanoleuca*: [5,13]; eastern black-and-white colobus monkeys: [8]; Asian elephants, *Elaphus maximus*: [19]].

Bears are commonly housed within facilities accredited by the Association of Zoos and Aquariums [20] but are prone to exhibiting stereotypic behaviors [13,14,21,22]. Pacing is the most frequently observed stereotypy in bear species under human care [5], which is thought to be the result of a combination of factors, including space restriction, monotony, and the inability to complete the idiosyncratic expression of some natural behaviors [4,5,23,24]. Several studies have demonstrated a reduction of aberrant behaviors through enrichment, but few have examined how access affects behavior. One study reported that offering a honey-filled log and scatter feeding reduced the pacing of a sloth bear (*Melursus ursinus*) and an American black bear (*U. americanus*) [7]. Various food items within ice blocks were found to increase activity levels and decrease the occurrence of stereotypic behaviors in a Kodiak bear (*U. arctos middendorffi*) and a polar bear (*U. maritimus*) [11]. While there is limited research examining how increasing access to a variety of locations impacts behavior among bears, one study found that polar bears with access to off-habitat areas reduced engagement in stereotypy and increased time spent in species-typical behaviors [14]. In another study, giant pandas (*A. melanoleuca*) exhibited a reduction in the occurrence of stress-related behaviors when given the free choice of habitat location [13].

Our study aims to assess how greater choice, in the form of access to multiple areas within a habitat, effects the welfare of American black bears. We quantified and compared behaviors exhibited by two American black bears housed at the North Carolina Zoo across days where access to alternate spaces was limited and those where access was available. We predicted that giving the bears choice and control over their space use via access to various habitats would reduce stereotypic behaviors (pacing) and increase species-specific behaviors (foraging). While some level of stress is expected and species-appropriate, we are focused on stereotypic patterns, specifically pacing, which are considered aberrant and undesired [1,4]. Because visibility for the guest view is a common concern across zoos [25–28], we also assessed the effect of access on guest visibility. We believe that this study will supplement our understanding of how to promote good welfare among zoo-housed carnivores and inspire further research into the effects of choice and control.

## 2. Materials and Methods

### 2.1. Study Subjects

The two American black bears in this study are sisters, Luna and Nova, housed together at the North Carolina Zoo in Asheboro, North Carolina. The bears were rescued from a private facility at 5 months old and moved to their current location, and at the time of the study, the bears were 12 years old. This research was approved by the Research Review Committee at the North Carolina Zoo.

### 2.2. Data Collection and Study Design

Data collection occurred from August 2020 through January 2021. Observations took place Monday through Friday between 9 a.m. and 5 p.m.

The North Carolina Zoo's black bear enclosure includes the habitat which is within guest view and the holding area, concrete yard, and chute which are outside of guest view (Figure 1). The habitat features a naturalistic design with a pool, rock structures, dens, grass, and trees. The guest-view habitat is approximately 3/4 acre, the indoor stalls are each around $3 \times 3$ m and the off-guest-view yard is $9 \times 9$ m. The areas out of guest view (holding area, concrete yard, and chute) include fewer natural features, but additional space. To assess the effects of access on behavior, three days per week (Monday, Wednesday, and Friday), the bears were given "full access" to all areas. Two days per week (Tuesday and Thursday), the bears were given "limited access", where the bears were not given access to the areas out of guest view, from 9 a.m. until 4 p.m. This schedule allowed the animal caretakers to reliably follow our protocol and control all other aspects of care during

the week. Generally, the daily management of the bears was consistent across the weeks: the bears were taken off of the habitat to a holding area from 8–9 a.m. while the habitat was cleaned and food and enrichment were scattered, then allowed back out on their habitat. Additional food was thrown onto the habitat during the mid-day (11–12 p.m.) and then later in the afternoon (3:30–4:30 p.m.) the bears were again taken to holding while food was scattered on their habitat. The foods and enrichment used were kept consistent throughout each week. While no data were collected on the weekends, we should note that access was limited over these days.

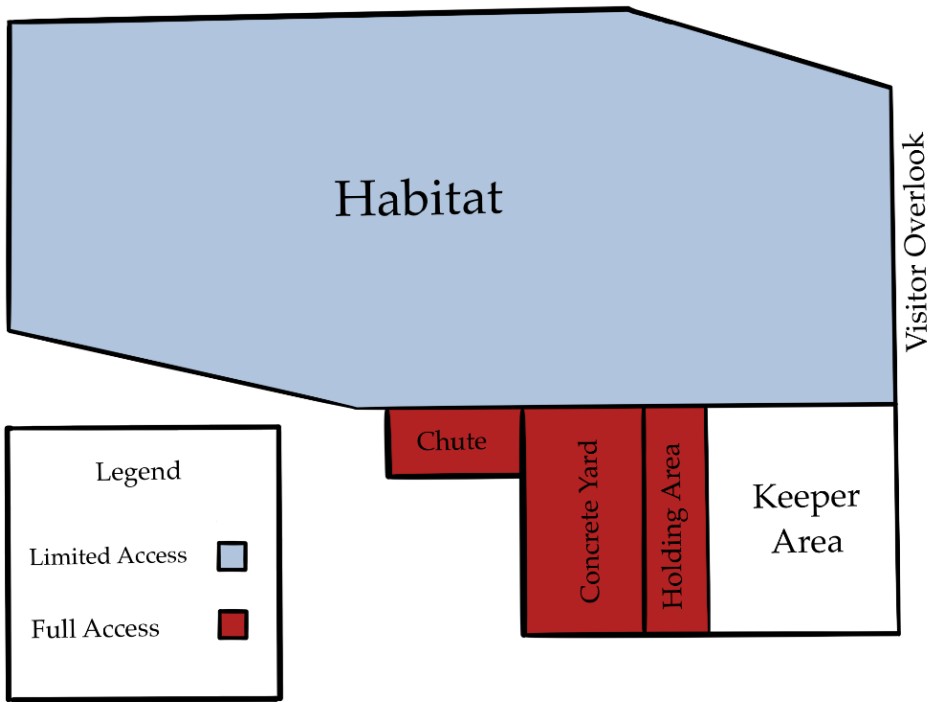

**Figure 1.** Map of black bear enclosure at the North Carolina Zoo.

An ethogram was developed based on previous studies on bears and behaviors were grouped into the following categories: active, social, investigative, maintenance, and self-directed (Table 1) [23,29,30]. Data were recorded using ZooMonitor software (version 3.2), with two-minute interval scan sampling within twenty-minute sessions [31,32]. Five hours of observation sessions were collected per day (mean 15 sessions per day). Observations were recorded by a single observer. In addition, intervals where the bears were recorded as "out of sight" were removed, as the behavior could not be determined [33,34]. In total, 2400 intervals were recorded for Luna and 2872 intervals were recorded for Nova.

**Table 1.** Ethogram of black bear behavior. Behaviors have been grouped by their function into five classes.

| Category | Behavior | Description |
|---|---|---|
| Active | | |
| | Pace | Individual walks repetitively back and forth along a fixed path for at least two repetitions (A-B-A) |
| | Walk | Individual walks from one area to another |
| | Active Rest | Individual is sitting or lying while remaining alert with eyes open and head raised |
| | Inactive Rest | Individual is sitting or lying and is immobile or sleeping |

**Table 1.** *Cont.*

| Category | Behavior | Description |
|---|---|---|
| Social | | |
| | Play | Individual exhibits "play" behavior with conspecific, can include chasing, wrestling, playful biting, embracing, following in play, object play, etc. |
| | Affiliative | Individual interaction with conspecific that does not involve play or agonism |
| | Agonism | Individual interaction with conspecific that does not include play or affiliation, can include swatting, charging, stalking, etc. |
| | Displacement | Individual moves for a conspecific or takes the spot of a conspecific |
| Investigative | | |
| | Dig | Individual manipulates substrate with paws for reasons other than acquiring food |
| | Interaction with non-conspecifics | Individual interacts with an animal in the enclosure that is not a conspecific |
| | Interaction with environment | Individual interacts with the environment by sniffing or manipulating objects, buildings or substrates |
| | Human-directed | Individual directs all of their attention to a human by staring, vocalizing, etc. |
| Maintenance | | |
| | Eat | Individual is actively foraging for food |
| | Drinking | Individual drinks water |
| Self-Directed | | |
| | Object rub | Individual rubs body on an object |
| | Self-groom | Individual licks, cleans, or grooms self |
| | Masturbating | Individual rubs genital area |
| Other | | |
| | Out of Sight | Individual is not within the observer's view |

### 2.3. Data Analysis

Data analyses were performed in RStudio version 4.1.2 [35]. The models considered each bear individually. Pacing, foraging, and visibility models were fitted as generalized linear mixed models (GLMMs) using the packages stats and lme4 [31,32]. The Akaike information criterion (AIC) was used for model selection with the package aiccmodavg [33,34,36], where the model with the lowest AIC score (by at least 2) was considered the best fit. Models were plotted with the package ggplot2 [37].

### 2.4. Pacing

Our first model was designed to explore the likelihood of stereotypic pacing as predicted by access. Here, the dependent variable was binomial, representing if the bear was observed to pace within an interval (Luna: pace observed, N = 353, pace not observed, N = 2047; Nova: pace observed, N = 211, pace not observed, N = 2661). The full model considered "access" as a fixed effect (binomial; Luna: limited, N = 1236, full, N = 1164; Nova: limited, N = 1241, full, N = 1631). We also included a term "period of day" to account for behavioral changes over time, as husbandry routines may affect observed behaviors among animals under human care [21,38,39] (Luna: morning, N = 781, mid-day, N = 944, afternoon, N = 675; Nova: morning, N = 1015, mid-day, N = 1079, afternoon, N = 778). Because the season has been shown to influence stereotypy in bears [23,40,41], it was included as a random effect (Luna: denning, January–March, N = 440, non-denning, August–October, N = 1960; Nova: denning, N = 739, non-denning, N = 2133).

### 2.5. Foraging

Our second model tested the influence of access on foraging behaviors. The dependent variable was binomial, representing if the bear was observed to forage within an interval (Luna: foraging observed, N = 346, foraging not observed, N = 2054; Nova: foraging observed, N = 415, foraging not observed, N = 2457). The fixed effects used were the

same as in the previous model, and the season was included as a random effect due to its influence on foraging in bears [42–44].

*2.6. Visibility*

Our third model explored how visibility could be affected by access. The dependent variable was binomial, representing whether the bear was observable from a guest viewing location (Luna: visible, N = 2484, not visible, N = 1398; Nova: visible, N = 3456, not visible, N = 699). Full models included access as a fixed term (see sample sizes above). The period of day was, again, included and the season was included as a random effect (see sample sizes above).

**3. Results**

*3.1. Pacing*

3.1.1. Luna

We found a significant effect of both access and period of day on the likelihood of Luna's pacing (Table 2, Figure 2). Full access decreased the likelihood of pacing by 21%. The probability of pacing was highest in the afternoon at 32%. This decreased by 10% at mid-day and 23% in the morning. The season was fitted as a random effect ($\sigma^2$ (variance) = 0.38).

**Table 2.** Final model results for GLMM exploring the effects of access and period of day on the likelihood of Luna's pacing. The term "period of day" was releveled to explore combinations of the three-level term.

| Term | Estimate | SE | *p*-Value |
|---|---|---|---|
| Intercept | −0.16 | 0.46 | 0.73 |
| Full vs. limited access | −1.47 | 0.19 | <0.001 |
| Period of day: morning vs. afternoon | −1.73 | 0.18 | <0.001 |
| Period of day: afternoon vs. mid-day | 0.59 | 0.13 | <0.001 |
| Period of day: mid-day vs. morning | 1.14 | 0.18 | <0.001 |

3.1.2. Nova

Again, we found a significant effect of access and period of day on Nova's probability of pacing (Table 3, Figure 2). Full access reduced Nova's likelihood of pacing by 6%. The probability of pacing decreased throughout the day. In the afternoon the probability was 11%, which decreased to 8% in the mid-day and 10% in the morning. The season was fitted as a random effect ($\sigma^2$ = 11.63).

**Table 3.** Final model results for GLMM exploring the effects of access and period of day on the likelihood of Nova's pacing. The term period of day was releveled to explore combinations of the three-level term.

| Term | Estimate | SE | *p*-Value |
|---|---|---|---|
| Intercept | −3.86 | 3.30 | 0.24 |
| Full vs. limited access | −1.54 | 0.20 | <0.001 |
| Period of day: morning vs. afternoon | −3.07 | 0.31 | <0.001 |
| Period of day: afternoon vs. mid-day | 1.56 | 0.20 | <0.001 |
| Period of day: mid-day vs. morning | 1.51 | 0.32 | <0.001 |

*3.2. Foraging*

3.2.1. Luna

We found that access had a significant effect on foraging (Table 4, Figure 3). When given full access, the likelihood of Luna engaging in foraging behaviors increased by 8%. The season was fitted as a random effect ($\sigma^2$ = 0.26).

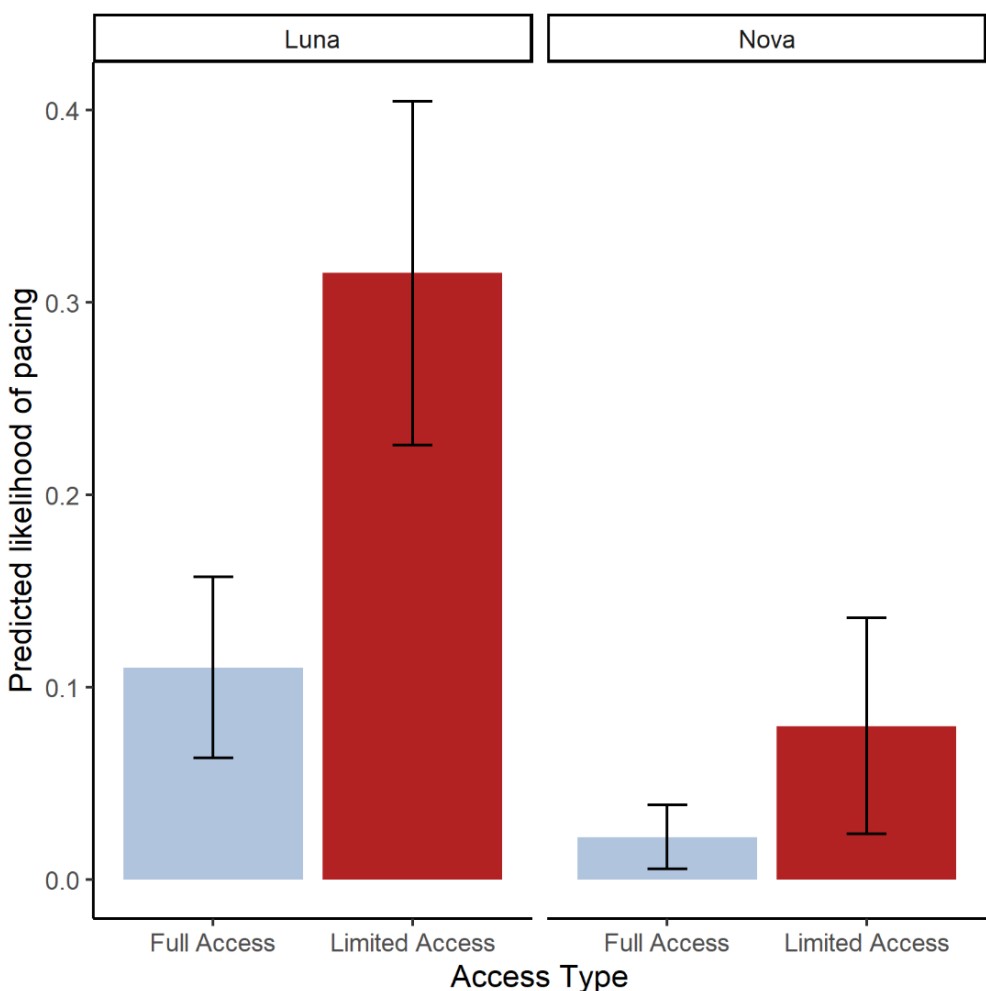

**Figure 2.** Predicted likelihood of pacing by access type. Error bars represent 95% standard error.

**Table 4.** Final model results for GLMM exploring the effects of access and period of day on the likelihood of Luna's foraging. The term period of day was releveled to explore combinations of the three-level term. Terms removed from the final model are shown below the line.

| Term | Estimate | SE | *p*-Value |
|---|---|---|---|
| Intercept | −2.53 | 0.38 | <0.001 |
| Full vs. limited access | 0.79 | 0.13 | <0.001 |
| Period of day: morning vs. afternoon | −0.23 | 0.15 | 0.119 |
| Period of day: afternoon vs. mid-day | 0.23 | 0.15 | 0.119 |
| Period of day: mid-day vs. morning | −0.10 | 0.14 | 0.480 |

### 3.2.2. Nova

We found a significant effect of both access and period of day on Nova's likelihood of foraging (Table 5, Figure 3). With full access, the likelihood of foraging increased by 3%. The probability of foraging was highest in the afternoon and decreased throughout the day. In the afternoon, the likelihood was 14%; this decreased by 2% in the mid-day and 5% in the morning. The season was fitted as a random effect ($\sigma^2 = 0.30$).

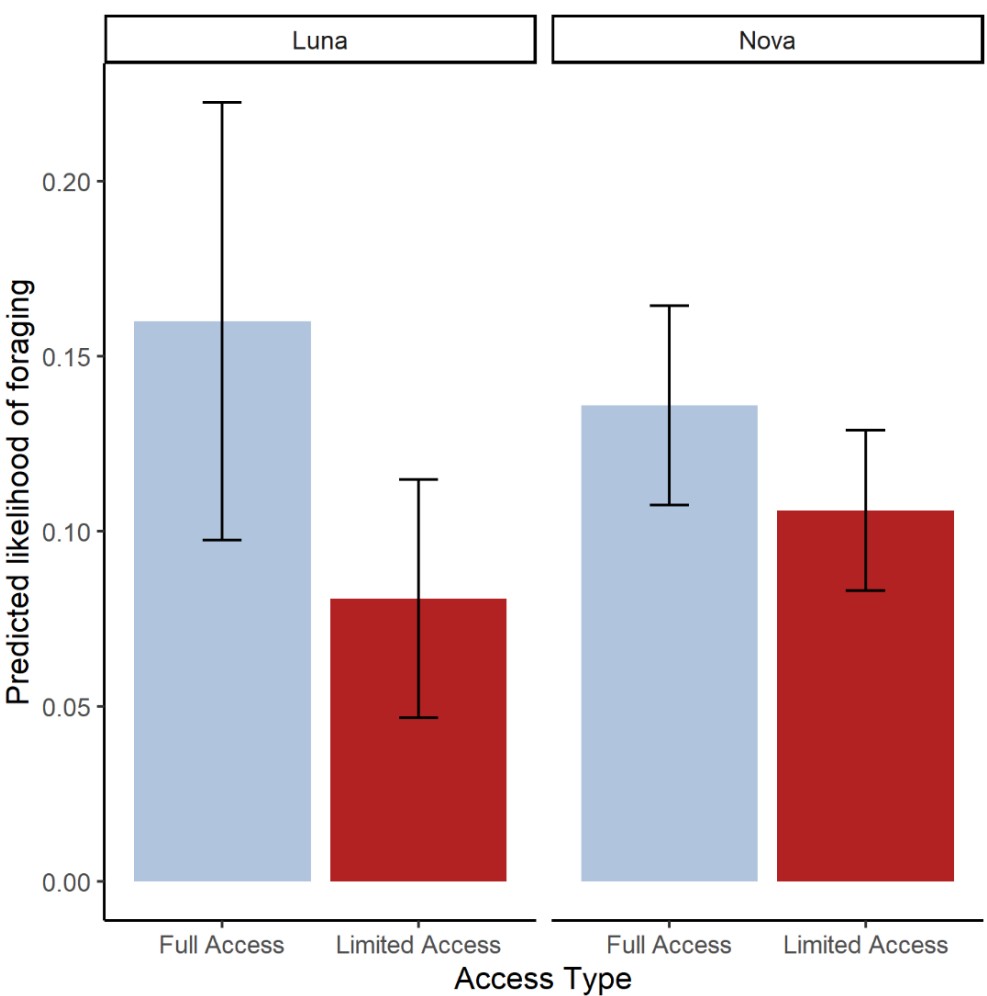

**Figure 3.** Predicted likelihood of foraging by access. Error bars represent 95% standard error.

**Table 5.** Final model results for GLMM exploring the effects of access and period of day on the likelihood of Nova's foraging. The term period of day was releveled to explore combinations of the three-level term.

| Term | Estimate | SE | *p*-Value |
|---|---|---|---|
| Intercept | −2.05 | 0.41 | <0.001 |
| Full vs. limited access | 0.29 | 0.12 | <0.05 |
| Period of day: morning vs. afternoon | −0.49 | 0.13 | <0.001 |
| Period of day: afternoon vs. mid-day | 0.16 | 0.13 | 0.20 |
| Period of day: mid-day vs. morning | 0.32 | 0.13 | <0.05 |

*3.3. Visibility*

3.3.1. Luna

We found that full access significantly decreased Luna's chances of being visible to visitors (Table 6, Figure 4). When given full access, the likelihood of Luna being visible in the habitat decreased by 57% compared to when she had limited access.

**Table 6.** Final model results for GLMM exploring the effect of access on the likelihood of Luna's visibility. Terms rejected in the final model (determined using AIC) are shown below the line.

| Term | Estimate | SE | *p*-Value |
|---|---|---|---|
| Intercept | 2.69 | 0.24 | <0.001 |
| Full vs. limited access | −3.27 | 0.14 | <0.001 |
| Period of day: morning vs. afternoon | 0.01 | 0.12 | 0.94 |
| Period of day: afternoon vs. mid-day | 0.13 | 0.12 | 0.28 |
| Period of day: mid-day vs. morning | −0.12 | 0.12 | 0.30 |

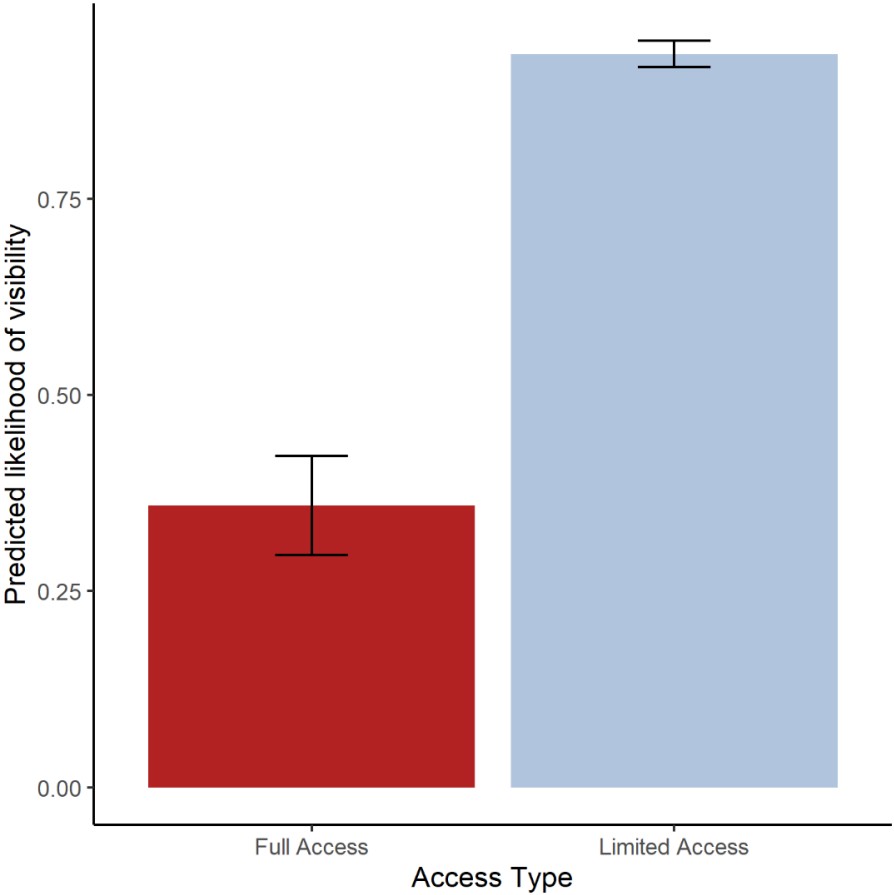

**Figure 4.** Predicted likelihood of Luna's visibility across different types of access. Error bars represent 95% standard error.

### 3.3.2. Nova

We found that the period of day had a significant effect on the likelihood of Nova's visibility to visitors (Table 7, Figure 5). Nova's probability of being visible in the habitat was highest in the morning at 86% and decreased by 1% at mid-day and 10% in the afternoon. Nova's visibility in the habitat was not significantly impacted by access.

**Table 7.** Final model results for GLMM exploring the effect of time of day on the likelihood of Nova's visibility. Terms rejected in the final model (determined using AIC) are shown below the line.

| Term | Estimate | SE | *p*-Value |
|---|---|---|---|
| Intercept | 1.19 | 0.30 | <0.001 |
| Period of day: morning vs. afternoon | 0.72 | 0.13 | <0.001 |
| Period of day: afternoon vs. mid-day | 0.56 | 0.12 | <0.001 |
| Period of day: mid-day vs. morning | 0.16 | 0.13 | 0.22 |
| Full vs. limited access | −18.56 | 478.66 | 0.97 |

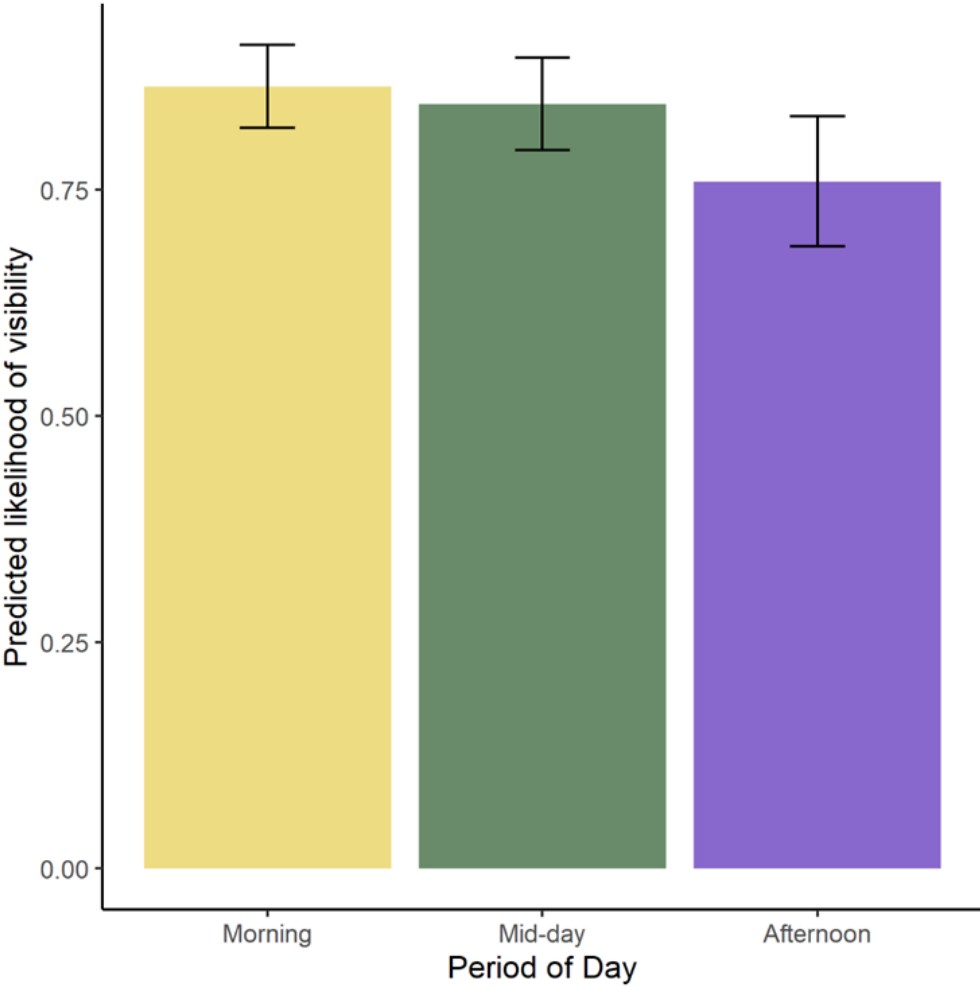

**Figure 5.** Predicted likelihood of Nova's visibility across different types of access. Error bars represent 95% standard error.

## 4. Discussion

We report that access to multiple areas within a habitat significantly reduced stereotypy and increased foraging among two zoo-housed black bears. In addition, full access only reduced the likelihood of visibility for one bear, while the other remained in guest view, suggesting that this method of enrichment may not always affect the guest experience. Overall, this case study provides evidence that environmental enrichment in the form of access improves welfare while having a limited impact on visibility to visitors.

This case study builds on the current understanding of the efficacy of providing choice and control to animals under human management [13,14,18,22]. In particular, choice and control over space use was found to improve welfare, even when the additional space was not utilized or highly enriched [13,14,18,22]. When a pair of male and female sibling polar bears were given the choice to access indoor dens, pacing was reduced, and social play increased [14]. For carnivores and other species that require larger home range sizes, artificial housing can constrain locomotion, which may be associated with an increase in stereotypical behaviors [45–47]. Providing access to more spaces, even if unused, can serve as a form of choice and control for the animal, enabling options for space use [13,14,21]. The benefits of providing access can lead to a decrease in stress and time spent in stereotypic behaviors, allowing for more time to be spent performing species-typical behaviors. [10,14,22,48].

It should be noted that time of day was found to have a significant effect on behavior. We believe that a few factors may have contributed to the increase in Nova's foraging

activity in the mid-day and afternoon. Guest counts wane towards the end of the day and that may reduce the occurrence of stress-related behavior. The bears are fed during mid-day so we can assume that they forage more in the afternoon. An increase in pacing is likely due to husbandry scheduling and routines, where the anticipation of feedings and keeper interactions can influence behavior [21,38,39,44]. Our findings are consistent with previous studies on carnivores, demonstrating that stereotypic behaviors and activity are closely tied to the time of day due to routine husbandry schedules [21,38,49]. Anticipatory pacing may occur before an expected feeding. Due to feeding occurring over a two-hour period, a distinction between pre-feeding anticipatory behavior and stereotypic behavior was not made; however, if possible, this distinction should be noted [50]. We believe that the visibility was highest in the morning due to husbandry routines. The keepers service the habitat as early as possible and leave diet and enrichment around the habitat. The bears are more likely to search for enrichment and diet following this. Our paper further highlights the importance of considering the time of day when monitoring welfare and the effects of husbandry changes.

Finally, we explored the guest experience, as it relates to access. Because visitor density and the duration of time spent observing a habitat generally increases when animals are visible and active [42,51], animal visibility is a primary concern across zoological institutions [25–28]. We report that full access significantly decreased pacing behavior for both bears, but only affected the visibility of one bear, Luna. However, it should be noted that among the two bears, at least one was visible in the habitat 74% of all the observed time, when given full access. As visitor perception of a zoo decreases when stereotypical behaviors are observed [43], beyond welfare and ethical concerns, it is useful to the success of an institution to mitigate stereotypy from the guest perspective. With the reduction of stereotypic behavior and a minimal effect on visibility, our study suggests the value of providing access to both the individual animal and the guest.

Two important caveats are notable in this study. First, our study was limited to two focal animals. Future research would benefit from exploring the effects of access on various behaviors across carnivores or various ages and sexes.

Second, given the constraints of the daily routine for animal caretakers, we implemented a consistent schedule for limited and full access days. However, staff ensured management procedures, including feeding times, training schedules, habitat servicing routines, and so on, were consistent across days of the week. In addition, guest numbers do not often fluctuate during weekdays, so visitor numbers are not expected to have had an impact. Therefore, this consistent schedule is unlikely to have affected our results. Future work, though, may benefit from exploring this possibility.

Overall, our study suggests that access can reduce stereotypic behaviors (pacing) and increase desired behaviors (foraging) without fully compromising visibility for zoo visitors. Choice and control are more likely responsible for these outcomes than enrichment and stimulus diversity since the added areas of access were not more enriched than the habitat. The individual bears had varying degrees of change in response to access, which we believe are due to personality differences. Building habitats in zoos is very time-consuming and expensive. Opening gates to areas that are already built presents a no cost and simple method to improve welfare under human care. This case study can serve as the foundation for future research to explore the value of choice and control as a method to enrich the lives of zoo-housed carnivores.

**Author Contributions:** Conceptualization, K.B. and E.L.; methodology, K.B.; software, K.B.; validation, K.B., C.H. and E.L.; formal analysis, K.B.; investigation, K.B.; resources, E.L.; data curation, K.B.; writing—original draft preparation, K.B. and C.H.; writing—review and editing, E.L.; visualization, K.B.; supervision, E.L.; project administration, E.L. All authors have read and agreed to the published version of the manuscript.

**Funding:** This research received no external funding.

**Institutional Review Board Statement:** Not applicable.

**Informed Consent Statement:** Not applicable.

**Data Availability Statement:** The data presented in this study are available on request from the corresponding author. Code associated with this study are available at https://github.com/kelbru/ncblackbears.git (accessed on 23 January 2023).

**Acknowledgments:** We are grateful for the support of the North Carolina Zoo's Northwoods team who assisted with implementing this study. In particular, we thank Erin Ivory, Chris Lasher, Natalie Johnson, Curtis Malott, Guy Burlingame, and Ashley Rydzfski. We also thank Corinne Kendall and Hunter Bunting for conducting the pilot work for this project.

**Conflicts of Interest:** The authors declare no conflict of interest.

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
