# Peer review of "Access to Multiple Habitats Improves Welfare: A Case Study of Two Zoo-Housed Black Bears (Ursus americanus)"

_2673-5636, doi:10.3390/jzbg4010010_

Round 1

Reviewer 2 Report

Questions and comments:

1.      The authors contrast species-specific behaviors with stress-related behavior. However, there are a lot of natural behaviors that naturally come with high levels of stress (such as animal behavior during breeding season). The use of the term requires clarification.

2.       Sometimes black bears exhibit oral stereotypies in a captive environment. Please, specify whether other forms of stereotypic behavior were observed besides pacing during this study.

Bears in captivity can exhibit self-directed behavior toward different parts of their bodies to attract visitors. In the ethogram, there is a “Human-directed” behavior when the bear is staring toward humans. Were there any looks in the direction of visitors during the “Masturbating” behavior and were there any self-scratching of other parts of the body?

3.      Specify how many times a day and at what time the animals were fed. It is also not clear what kind of enrichment the bears had besides structural and social. Feeding/enrichment time may have influenced the level of pacing before feeding. Were these factors controlled (feeding/enrichment) during the two phases of observation?

4.      How many observation sessions were during the day and were they randomized and/or evenly distributed throughout the day? Have two bears been observed at the same time? How many observers collected data? If there were more than one observer does the agreement between observers was reached?

5.      Clarify why you write that “intervals where the bears were recorded as “out of sight” were removed…”, but then analyzed the "Visibility", which does not reach 100%. If “not visible” was recorded, please add it to the ethogram.

6.      In the Material and Method section, it was described that the unit of analysis was the 2-minute interval. In such a data aggregation, strong data autocorrelation can be obtained. Usually, to deal with it, a session is used as a unit of analysis (a 20-minute session in your case). This allows one to deal with autocorrelation, which greatly affects the result of linear models.

7.      In the Results, there are tables of the influence of the 'access' and 'period of day' factors on different forms of bears' behavior are given. However, it remains unclear whether the interaction between these factors has been taken into account. For example, in Table 2, the equivalent contribution of these factors is observed. But the authors discuss them separately, which does not allow a more accurate idea of the contribution of each factor and what actually influenced the behavior of animals during the two phases of observation. The interaction of these factors is also not presented in the discussion.

Abstract

Line 11. Please, specify the “positive welfare” of animals.

Line 11-12. It is better to say “various/different forms of enrichment” instead of “forms of enrichment”.

Line 13. Please, replace here and further “stress behaviors” with “stress-related behavior”, since “stress behaviors” is slang.

Introduction

Line 29-30. It is necessary to clearly say what is positive and negative welfare. In general, welfare status is related to an individual state, not just to its behavior. A researcher can try to measure the current welfare status, using different welfare indicators that are on a scale from positive to negative. It is proposed to correct that animals in the positive state can exhibit different forms of natural behavior, while in the negative state, an animal is perform stress-related behaviors, like pacing, etc.

Line 31-32. In the sentence “Stereotypic behaviors can disrupt engagement in species-specific behaviors”, the cause-and-effect relationship is lost. Stereotypic behavior cannot disrupt engagement in natural behaviors. It is better to say about the negative state of the animal, in which the individual exhibits stereotypical behavior. Stereotypical behavior is most likely due to the lack of choice and control in the environment. Therefore, an animal cannot exhibit species-specific behaviors due to its mood/state under suboptimal conditions.

Line 34. It is better to conclude that all these efforts (“enrichment, exhibit additions, and alteration in husbandry routines”) are related to choice and control, not just mental stimulation;  that can help to link with the next paragraph.

Line 43. In the cited works [13–17], stress hormones were not measured everywhere. If you are talking about stress-related behavior, and not stress hormones, then the phrase “reduce stress” needs to be changed.

Line 63-65. The paper about choice for pandas can be added here that was cited earlier: “Owen, M.A.; Swaisgood, R.R.; Czekala, N.M.; Lindburg, D.G. Enclosure Choice and Well-Being in Giant Pandas: Is It All about Control? Zoo Biol. 2005, 24, 475–481, doi:10.1002/zoo.20064”

Line 75-79. Probably this paragraph should be moved to the discussion or conclusion section.

In the introduction or discussion, you can add a link to an article about the stereotypical behavior of the black bear: Vickery, S., & Mason, G. (2004). Stereotypic behavior in Asiatic black and Malayan sun bears. Zoo Biology: Published in affiliation with the American Zoo and Aquarium Association, 23(5), 409-430.

Material and Methods

Line 82-85.  Add the clarification that the bears shared the same enclosure.

Line 90. Can you add the sizes of the enclosure/habitat area and size areas out of the guest view area?

Figure 1. The diagram shows only chute A, it is better to leave just chute.

It is not entirely clear what the colored borders indicate, which are offset from the enclosure scheme. Several zones are also not marked with labels. Please, add a legend and label the rest of the zones.

Line 94-98. It is necessary to clarify what access to the areas of the enclosures was on weekends because this feature can also affect the results.

Table 1. Two styles of describing behavior are presented, with or without reference to the individual. It is necessary to keep one style for behavior description.

The definition of the term “Masturbating” includes elements of anthropomorphism (“pleasuring self”), you need to change this part.

Data Analysis

Specify the name of the package that found the smallest AIC (e.g., library MuMIn).

The data has been imbalanced. Clarify how you dealt with class imbalance when building models (e.g., SMOTE, class weight, upsampling method).

Specify the plotting library.

Results

In Figures 2-6, you need to remove the grid and increase the resolution.

You can combine figures for one form of behavior for two animals.

Line 150. Please, specify the symbol “σ2” (variance).

Figures 4-5 have the wrong Y-axis label.

Figure 7 is not presented in the paper, but there is a link to it.

Discussion

The first sentence of the first paragraph and the first sentence of the second paragraph of the discussion almost completely duplicate each other (“significantly reduced stereotypy and increased foraging among two zoo-housed black bears.” and “Our study found that full access reduced overall pacing and increased time spent foraging in zoo housed black bears”). Duplication does not add new information.

Line 223-225. Please add more recent papers about the link between pacing and the size of home range, e.g. Kroshko, J., Clubb, R., Harper, L., Mellor, E., Moehrenschlager, A., & Mason, G. (2016). Stereotypic route tracing in captive Carnivora is predicted by species-typical home range sizes and hunting styles. Animal Behaviour, 117, 197-209.

Line 226-227. The phrase “create an illusion of additional areas to range and explore, mimicking natural home ranges” needs clarification. Having a small off-exhibit area without enrichment is unlikely to mimic natural home ranges. If you are talking about providing an additional opportunity to space, then this item is not needed and it is enough to leave the second item about choice and control.

Line 231-237. It is necessary to distinguish between anticipatory behavior that is performed prior to feeding and stereotypic behavior: (Watters, J. V., Krebs, B. L., & Eschmann, C. L. (2021). Assessing animal welfare with behavior: Onward with caution. Journal of Zoological and Botanical Gardens, 2(1), 75-87.).

According to the results of this paper,  pacing was more commonly observed in the afternoon. But also showed throughout the day. Therefore, a statement can be added about the time of the feeding and pacing in the discussion.

Line 241. There is an extra word “because” and no dot after the previous sentence.

References

44. The first letters in the name of the journal “anim welf” should be written in upper case. 
